# Functional Annotation of *Lactiplantibacillus plantarum* 13-3 as a Potential Starter Probiotic Involved in the Food Safety of Fermented Products

**DOI:** 10.3390/molecules27175399

**Published:** 2022-08-24

**Authors:** Tariq Aziz, Muhammad Naveed, Abid Sarwar, Syeda Izma Makhdoom, Muhammad Saad Mughal, Urooj Ali, Zhennai Yang, Muhammad Shahzad, Manal Y. Sameeh, Mashael W. Alruways, Anas S. Dablool, Abdulraheem Ali Almalki, Abdulhakeem S. Alamri, Majid Alhomrani

**Affiliations:** 1Beijing Advanced Innovation Center for Food Nutrition and Human Health, Beijing Engineering and Technology Research Center of Food Additives, Beijing Technology and Business University, Beijing 102401, China; 2Pak-Austria Fachhochschule—Institute of Applied Sciences and Technology, Mang, Haripur 22621, Pakistan; 3Department of Biotechnology, Faculty of Life Sciences, University of Central Punjab, Lahore 54590, Pakistan; 4Institute of Basic Medical Sciences, Khyber Medical University, Peshawar 25124, Pakistan; 5Chemistry Department, Faculty of Applied Sciences, Al-Leith University College, Umm Al-Qura University, Makkah 24831, Saudi Arabia; 6Department of Clinical Laboratory Sciences, College of Applied Medical Sciences, Shaqra University, Shaqra 15273, Saudi Arabia; 7Department of Public Health, Health Sciences College Al-Leith, Umm Al-Qura University, Makkah al-Mukarramah 24382, Saudi Arabia; 8Department of Clinical Laboratory Sciences, The Faculty of Applied Medical Sciences, Taif University, P.O. Box 11099, Taif 21944, Saudi Arabia

**Keywords:** *Lactiplantibacillus plantarum* 13-3, functional annotation, probiotic, starter culture, whole genome, food safety

## Abstract

The important role of *Lactiplantibacillus plantarum* strains in improving the human mucosal and systemic immunity, preventing non-steroidal anti-provocative drug-induced reduction in T-regulatory cells, and as probiotic starter cultures in food processing has motivated in-depth molecular and genomic research of these strains. The current study, building on this research concept, reveals the importance of *Lactiplantibacillus plantarum* 13-3 as a potential probiotic and bacteriocin-producing strain that helps in improving the condition of the human digestive system and thus enhances the immunity of the living beings via various extracellular proteins and exopolysaccharides. We have assessed the stability and quality of the *L. plantarum* 13-3 genome through de novo assembly and annotation through FAST-QC and RAST, respectively. The probiotic-producing components, secondary metabolites, phage prediction sites, pathogenicity and carbohydrate-producing enzymes in the genome of *L. plantarum* 13-3 have also been analyzed computationally. This study reveals that *L. plantarum* 13-3 is nonpathogenic with 218 subsystems and 32,918 qualities and five classes of sugars with several important functions. Two phage hit sites have been identified in the strain. Cyclic lactone autoinducer, terpenes, T3PKS, and RiPP-like gene clusters have also been identified in the strain evidencing its role in food processing. Combined, the non-pathogenicity and the food-processing ability of this strain have rendered this strain industrially important. The subsystem and qualities characterization provides a starting point to investigate the strain’s healthcare-related applications as well.

## 1. Introduction

A main component of the human diet for centuries is fermented food [1]. In addition to the fact that those fermented foods derived from meat, milk, and plant foods have a longer shelf life as compared to fresh raw materials [2], this is also because of their higher water content and nutritional value. Plant- and animal-derived fermented foods plays a key role in the world food industry including Asia as well as Western countries and the Americas (Western hemisphere) [3]. These fermented foods contain nutrients with great potential in maintaining health and preventing diseases but also undergo changes in taste, texture, toxicity, and cooking time [4]. Initially, lactic acid bacteria (LAB) were isolated from fermented foods [5], and they are the most suitable candidate for enhancing fermentation in terms of product safety and have the capability to spontaneously adapt well during the fermentation process [3]. Recently, a lot of consideration has been given to assessing the role of bacteria in human and animal health, specifically in terms of the gastrointestinal tract (GIT) and protection against several diseases. The human stomach-related framework contains roughly 400 different bacterial species, and its overflow varies between people. Among them, a few probiotic Lactobacillus species in particular, *Lactobacillus acidophilus (L.)* [6], *L. pentosus*, *L. brevis*, *L. lactis* [7], *L. amylovorus*, *L. casei*, *L. bulgaricus* [8], *L. fermentum*, *L. plantarum* and *L. rhamnosus*, explicitly produce extracellular proteins, exopolysaccharides, bacteriocins and lipoteichoic acids [9] which impact on the wellbeing and physiology of the host by communicating with the epithelial cells [10] and improve the host resistant framework [11]. LAB is known to inhibit diverse environments such as carbohydrate-rich food plants, mucosa and intestinal environments of human and animals [12]. LAB isolated from fermented foods have the competency to help in digestive health (gut microbiome) are known as probiotics.

*Lactiplantibacillus plantarum* is a gastrointestinal bacterium which belongs to the family of *Lactobacillus* that lives in almost every type of environment [13]. It is a Gram-positive, anaerobic, and facultative bacteria. It can act as a food supplement for humans as well as for starters in animals [14]. This microorganism is particularly utilized in the fermentation of different foods [13]. Use of *L. plantarum* by humans has been documented from the start of fermentation almost a thousand ago [15]. In this use it is regarded as a safe microorganism [16]. *L. plantarum* is involved in balancing the microflora of the intestine of the human body [17], increasing the absorption of nutrients to the intestine and ultimately improving immunity [18]. This increased uptake can enhance the conversion rate of food into solid absorption of nutrients to the body [19]. It is not only useful for humans but also for the broilers because they maintain their normal flora [20]. It was also found that *L. plantarum* can survive the acidic or gastric environment of intestinal fluid and has a strong capacity to adhere to the walls of intestine [21]. As well as the aggregation capability of *L. plantarum* in the intestinal tract, it also contains the antagonistic activity against two species of bacteria such as *Salmonella typhi* and *Escherichia coli* [22]. This property of *L. plantarum* marks it as a probiotic organism [23]. Another study has shown evidence that *L. plantarum* along with *Aspergillus niger* can increase the weight gain of the body of the people above the age of 56 as a replacement for soya beans and cake made up of canola seeds [23]. This characteristic has demonstrated its potential as a probiotic in human food as well as in animal feeding [21].

Various dairy products are transporters via which consumers receive ample counts of probiotic *lactobacilli* [24,25]. Probiotic effects are determined by the number of viable microbial cells that reach the human gut [26]. Therefore, their viability in the product is considered vital to attain health effects. Probiotic lactobacilli are incorporated separately or in combination with other commercial culture into specific dairy products and several foods which have been reported, e.g., cornflakes, pomegranate juice, dough, cheese, fermented drink, yogurt, fermented milks, grape drink, soya milk chocolate, etc. [27,28,29]. Furthermore, they can also produce several useful metabolites during their growth and metabolism, for example, production of bioactive compounds, i.e., conjugated linoleic acid and other fatty acid metabolites [30,31,32,33,34,35].

Recently, the growing number of available genome sequences of *Lactiplantibacillus* strains has provided a better understanding of their genetic potential for probiotic properties and adaptability to environmental stresses. In our previous studies, *L. plantarum* YW11 strain isolated from Tibetan Kefir grains has demonstrated modulatory effects on gut dysbacteriosis [36], improving immune response and ameliorating inflammatory bowel disease (IBD) [37], as well as tolerance to acid and bile stress [38]. Similarly, we have also reported that *Lp* YW11 can be used as functional agent in the processing of fermented dairy products with enhanced textural stability and cholesterol-lowering, antioxidant, and antibiofilm bioactivities [39], and it has the capability of biotransformation of linoleic acid (LA) to conjugated linoleic acid (CLA) [33]. In addition to that we also reported that this strain *L. plantarum* 13-3 and 12-3 are also capable of converting LA to certain fatty acid metabolites [34,35]. However, the genetic basis of its probiotic characteristics and environmental adaptability is still mostly unknown.

## 2. Materials and Methods

### 2.1. Bacterial Strain and Culture Condition

The strain *Lactiplantibacillus plantarum* 13-3 (*L. plantarum* 13-3) was previously isolated from Tibetan Kefir grains in 2015 and was maintained as frozen (−9 °C) stocks in MRS broth supplemented with 20% (*v*/*v*) glycerol. Primarily, these strains were identified based on Gram reaction, catalase tests and cell morphology. Strains level identification was performed using API 50 CHL test (bio-Merieux, Marcy-l’Étoile, France) and 16S rDNA sequencing analysis as previously described by (Aziz, T., et al., 2022; 2021; 2020a; 2020b and 2020c; Jian et al.; 2017 and 2020; Zhang et al., 2022 and 2020; Yunyun et al. 2018; Wang et al., 2015) [32,33,34,35,36,37,38,39,40,41].

### 2.2. DNA Extraction and Whole Genome Sequencing

The genomic DNA was extracted using Wizard^®^ Genomic DNA Purification Kit (Promega, Madison, WI, USA) and quantified by TBS-380 fluorometer (Turner Bio Systems Inc., Sunnyvale, CA, USA) with an insert size of 15 kb. High quality DNA (OD260/280 = 1.8~2.0, >20 ug) was used for further analysis. Illumina sequencing libraries were prepared from the sheared fragments using the NEXTflex™ Rapid DNA-Seq Kit. Illumina sequencing libraries were prepared from the sheared fragments using the NEXTflex™ Rapid DNA-Seq Kit. Briefly, 5′ prime ends were first end-repaired and phosphorylated. Next, the 3′ ends were A-tailed and ligated to sequencing adapters. The third step was to enrich the adapters-ligated products using PCR. The prepared libraries then were used for paired-end Illumina sequencing (2 × 150 bp) on an Illumina HiSeq X Ten platform. The whole genome sequence with the accession number (GCA_004028315.1) of the selected strain (*L. plantarum* 13-3) was performed using single molecule real-time (SMRT) technology and Illumina sequencing platforms [42]. The Illumina data was used to evaluate the complexity of the genome. According to the manufacturer’s protocol, the genomic DNA was isolated by using the Qiagen DNA extraction kit and the process was completed between March–May 2019.

### 2.3. Genomic Investigation

The assembled genome was analyzed for the quality assessments of read via FASTQC, accessed on 15 May 2022. The sequenced entire genome of *L. plantarum* 13-3 was explained computationally by investigating different parts of genome. The utilitarian comment of qualities associated with different cell and metabolic pathways were anticipated by rapid annotation of utilizing subsystem technology (RAST) (https://rast.nmpdr.org/, accessed on 19 May 2022). The prophage areas inside the genome of *L. plantarum* 13-3 were anticipated by PHASTER web server (https://phaster.ca/, accessed on 20 May 2022). Clustered Regularly Interspaced Short Palindromic Repeats were recognized by CRISPRFinder (https://bioinformaticshome.com/devices/DNA-succession investigation/portrayals/CRISPRFinder.html, accessed on 22 May 2022).

Further, the arrangement comparability search was performed by NCBI BLAST (https://blast.ncbi.nlm.nih.gov/Blast.cgi, accessed on 24 May 2022). The Transporter Classification Database (TCDB) was utilized on 20 May 2022 to examine possible carriers from *L. plantarum* 13-3 (https://tcdb.org/) which were oppressed for their flagging abilities by Signal Peptide (SignalP 5.0) (https://services.healthtech.dtu.dk/service.php?SignalP-5.0, accessed on 21 May 2022).

### 2.4. Functional Annotation

Sugar dynamic chemicals (CAZy), basically glycoside hydrolases (GHs), were anticipated by CAZy data set (http://www.cazy.org/, accessed on 26 May 2022) and were additionally clarified by DbCAN meta server (https://bcb.unl.edu/dbCAN2/, accessed on 26 May 2022). Identification of antibiotic resistance factors CARD (Comprehensive Antibiotic Resistance Database) (http://arpcard.mcmaster.ca/) were also performed on 27 May 2022 to predict any antibiotic resistant gene in the *L. plantarum* 13-3 genome. Virulence factors using VFDB, on 29 May 2022 (a virulence factor database, http://www.mgc.ac.cn/VFs/main.Htm), were also predicted in the genome of *L. plantarum* 13-3. The circular diagram was predicted using CGView on 1 June 2022 that shows the predicted antibiotic resistant and virulent genes. The cryptoscopic protein–protein associations of annotated gene-creating qualities were anticipated by STRING on 3 June 2022.

### 2.5. Annotation of Genes Involved in Food Safety

The bacterial pathogenicity was analyzed via PathogenFinder web server (http://cge.cbs.dtu.dk/services/PathogenFinder/, accessed on 5 June 2022), which predicted whether the *L. plantarum* 13-3 was a pathogen or not. Prediction of putative gene cluster coding bacteriocins and other bioactive compounds, the location of biosynthetic quality groups and stress-related genes was researched utilizing the BAGEL4 (http://bagel.molgenrug.nl/) between 7 and 9 June 2022. The qualities connected with the flexibility to pH, bile salt hydrolase, temperature, and assimilation were recovered from EggNOG comment results on 10 June 2022. The exopolysaccharide biosynthesis quality bunches were examined by antiSMASH, accessed on 12 June 2022, bacterial variant in the entire genome of *L. plantarum* 13-3 (https://antismash.secondarymetabolites.org/#!/begin).

## 3. Results

### 3.1. Quality Assessment of 13-3 Genome

The overall reads of the *L. plantarum* 13-3 genome were predicted to be of good quality. The total sequence was of 5277,378 Base Pairs with a CG-Content of 46% (Figure 1).

### 3.2. Genomic Annotation

The genome size is 2,991,504 base sets, with 44.9% GC content, 228 subsystems and 32,918 qualities present in the genome of *L. plantarum* 13-3. The qualities present in the genome of *L. plantarum* 13-3 were contained on 26% sub-framework and 74% non-subsystem inclusion qualities. The subsystem inclusion contained 751 qualities altogether, of which 715 qualities were portrayed and 36 were speculative. Likewise, a non-subsystem inclusion contained 2167 altogether, of which 1247 qualities were anticipated, and 920 qualities were speculative. The complete classifications of the subsystem were 1060, of which 223 were starches, 99 cofactors, vitamins, prosthetic groups, pigments, 48 cell dividers and containers, 38 harmfulness, infection and guard, 5 potassium digestion, 14 random, 5 phage and plasmid parts, 34 film transport frameworks, 5 iron securing and digestion frameworks, 39 RNA digestion, 87 nucleosides and nucleotide frameworks, 133 protein digestion, 4 cell division and cell cycle, 15 guideline and cell flagging, 4 optional digestion, 48 DNA digestion, 32 unsaturated fat biosynthesis, 6 torpidity and sporulation, 16 breath, 21 pressure reaction, 168 amino corrosive subsidiaries, 3 Sulfur digestion, and 12 phosphorus digestion (Figure 2). The highlights of subsystem alongside their capacities are portrayed in the Appendix A.

The KEGG pathways of entire genome for the blend of fundamental biomolecules and optional metabolites particularly exopolysaccharides (EPS) are displayed in Figure 3.

### 3.3. Phage Site Prediction

PHASTER recognized two significant phage hits that will upgrade the hereditary variety and access the genomic variety during the bacterial development. The two significant hits were against infection and other bacterial species that were recently recognized for phage locales (Appendix A).

The graphical portrayal of the phage districts distinguished by PHASTER in the entire genome of *L. plantarum* 13-3 against two significant hits is displayed in Figure 4. No clustered routinely interspaced short palindromic rehashes (CRISPRs) were found in the *L. plantarum* 13-3 genome by CRISPRFinder.

### 3.4. Understanding of Transporter Proteins

The *L. plantarum* 13-3 genome was dissected for carrier proteins of which 39 proteins were distinguished as potential carriers that might be useful in different flagging pathways (Appendix A). The motioning of carrier proteins was examined by observation of sign peptides. The commonness of sign peptides was anticipated by TAT, LIPO and CS signals given by unambiguous amino corrosive at a specific cleavage site (Appendix A).

### 3.5. Carbohydrate Enzyme Prediction

The carb dynamic proteins (CAZy) informational collection is used to look at the genomic, essential, and biochemical information on CAZy that degrade, modify, or make glycosidic bonds. The CAZy data set has anticipated five significant classes of sugars in the genome of *L. plantarum* 13-3, i.e., Glycoside hydrolases, glycosyl transferases, carb esterase, assistant chemicals and starch restricting modulars (Figure 5). The genome of *L. plantarum* 13-3 contains 90 CAZy qualities, of which the glycoside hydrolase family and glycosyltransferase group of catalysts contributes significantly with around 42 and 38 qualities separately, which shows that *L. plantarum* 13-3 assumes a significant role in areas of strength due to its movement and guideline of the resistant framework against different microorganisms (Appendix A). The recognized qualities for the coding of starch coding compound were additionally commented on by dbCAN which examined HMMER, DIAMOND and CGC districts in the entire genome of *L. plantarum* 13-3 (Appendix A).

### 3.6. Functional Annotation of Genome

The functional characters of the genome were found to be involved in various food safety processes (Appendix A). The visualization of the genome was performed using CG-Viewer showing all the functional components of the genome in a graphical representation (Figure 6).

### 3.7. Protein–Protein Interaction Network

The distinguished genes engaged in the creation of functional genes were examined for their working regarding food safety and upgraded improvement in probiotic activity in people (Appendix A). The gene ontology components of genome *L. plantarum* 13-3 are described in Appendix A. The string organization of distinguished genes in the creation of functional components of the genome is displayed in Figure 7.

### 3.8. Pathogenicity of L. plantarum 13-3 Strain

The results of Pathogen Finder showed that *L. plantarum* 13-3 has 0.19 probability of being a human pathogen, with a pathogenic coverage of 0.43%. This percentage shows that *L. plantarum* 13-3 is a non-pathogenic bacterium for the human body (Figure 8). The overall 13 nearest matches of *L. plantarum* 13-3 strain to non-pathogenic bacteria are described in Appendix A.

### 3.9. Prediction of Secondary Metabolites

The antiSMASH system anticipated four fundamental areas that produce bacteriocins involved in food safety in the genome of *L. plantarum* 13-3, i.e., locale 1 (cyclic lactone autoinducer) (Figure 9a), district 2 (terpene) (Figure 9b), district 3 (T3PKS) (Figure 9c), and area 4 (RiPP-like) (Figure 9d). The Appendix A shows the near bacteriocins and other secondary metabolites creating locales of *L. plantarum* 13-3 with different genomes of *L. plantarum*.

## 4. Discussion

For many years, LAB have been employed in agricultural and food processing. They are typically absorbed during the processing of milk, fish, vegetables, meat, and fruits, and are used for enhancing the texture and taste of bread, sausage rolls, and alcohol, inhibiting microbe-dependent food degradation and extending its lifespan. Several LAB species colonize the mouth and intestinal tract of humans, making them potential mucosal vaccinations. *L. plantarum* is perhaps the most ubiquitous *Lactobacillus* species/strain with beneficial characteristics and is often found in various cultured food items. In addition, *L. plantarum* is frequently used in fermentation technology and the preparation of fresh foods is “generally regarded as safe” (GRAS) and has QPS certification [17]. FAO and WHO stipulate that *L. plantarum* isolates must have a high capacity to live in the intestinal system and be a safe strain for human beings.

This study was focused on the functional annotation on the *L. plantarum* 13-3 strain of *Lb. plantarum* as the strain which has the potential of being used as a potential starter probiotic as shown in [34]. Several other strains of the specie have been examined for the same potential. For instance, Ba et al. [43], exploited the capacity of *L. plantarum* strain KACC 92189 as a starter culture for fermented, safe foods. Similarly, Yilmaz et al. [44] reviewed other strains of the bacteria with functional applications in the industrial fermentation processes. These included strains 423 (fermentation of rice and wheat bran), QZ227 (fermented wheat silage), 299v, Heal 19, 299, and Lp900 (white beans and cauliflower), BX62 (fresh-cut apples, P-8 (fermented milk), PL62 (kimchi) [45]. In one of our previous studies, *L. plantarum* YW11 was investigated as a potential candidate in exopolysaccharide production with the aim of looking for opportunities to modify or engineer this strain to refocus its capacities in the human body as well as steer similar strains as viable probiotics in food and its complex mechanisms of CLA conversion [34].

This study, focusing on *L. plantarum* 13-3, looks for the same opportunities by annotating the strain. We found that the genome size of 13_3 2,991,504 base sets, with 44.9% GC content, 228 subsystems and 32918 qualities. No CRISPR-Cas site was found in this genome, however, two unique phage hit sites were identified for genome modulation. The CAZy data set has anticipated five significant classes of sugars in the genome of *L. plantarum* 13-3, i.e., Glycoside hydrolases, glycosyl transferases, carb esterase, assistant chemicals and starch restricting modulars. One of the most important findings of this study was the non-pathogenicity of the strain to the host with a 0.19 probability of being a human pathogen, with a pathogenic coverage of 0.43%. Four regions in the *L. plantarum* 13-3 genome were identified to produce bacteriocins involved in food safety. These included the cyclic lactone autoinducer, terpenes, T3PKS, and RiPP-like.

The identification of these four categories of compounds led to the agreement that the *L. plantarum* 13-3 does indeed have the potential for being used as a food safety agent in terms of regulating quorum sensing, having a beneficial influence on host nervous and tissue repair systems, and the general healing process of the host. Our findings were in line with previous studies such as Mull et al. [46], who discussed that cyclic peptides, similarly to the cyclic lactone autoinducer peptide, govern critical pathways of signal transduction, further targeting the polysaccharide biosynthesis and sugar utilization enzymes. In another study, terpenes and the T3PKS gene cluster were identified in *Pseudovibrio* genus with encouraging potential toward producing novel bioactive compounds, playing a significant role in food processing along with other industrial applications [47].

In the present study, in accordance with the findings discussed and the works of Aziz et al., [31] on bio-molecular analysis of food derived *Lb. strains*, Aziz et al., on the production of linolenic acid analogues from *L. plantarum* 13-3 [35], and Devi and Halami [48] on the genetic modulation of *L. plantarum* strains, holds that the *L. plantarum* 13-3 of *L. plantarum* can be used as a starter culture for probiotic, yet safe food processes. Furthermore, its non-pathogenicity opens roads for a wide spectrum of human health-related research and applications requiring the services of bacteria.

## 5. Conclusions

The current study reveals the importance of *L. plantarum* 13-3 as a potential probiotic and bacteriocin-producing strain that helps in improving the condition of the human digestive system and thus enhances the immunity of living beings via various extracellular proteins and exopolysaccharides. In the present study, the stability and quality assessment of the *L. plantarum* 13-3 genome is performed by de novo assembly and annotation through FAST-QC and RAST, respectively. Further, the eminent probiotic-producing components, secondary metabolites, phage prediction sites, pathogenicity and carbohydrate-producing enzymes in the genome of *L. plantarum* 13-3 are analyzed computationally. This study demonstrated that *L. plantarum* 13-3 is nonpathogenic bacterium with 218 subsystems and 32,918 qualities and five classes of sugars with several important functions. Two phage hit sites have been identified in the strain. Cyclic lactone autoinducer, terpenes, T3PKS, and RiPP-like gene clusters have also been identified in the strain, evidencing its role in food processing. Combined, the non-pathogenicity and the food-processing ability of *L. plantarum* 13-3 have rendered this strain industrially important. The subsystem and quality characterization provides a starting point to investigating the strain’s healthcare-related applications, which may also help researchers to discuss the potential health beneficial properties.

## Figures and Tables

**Figure 1 molecules-27-05399-f001:**
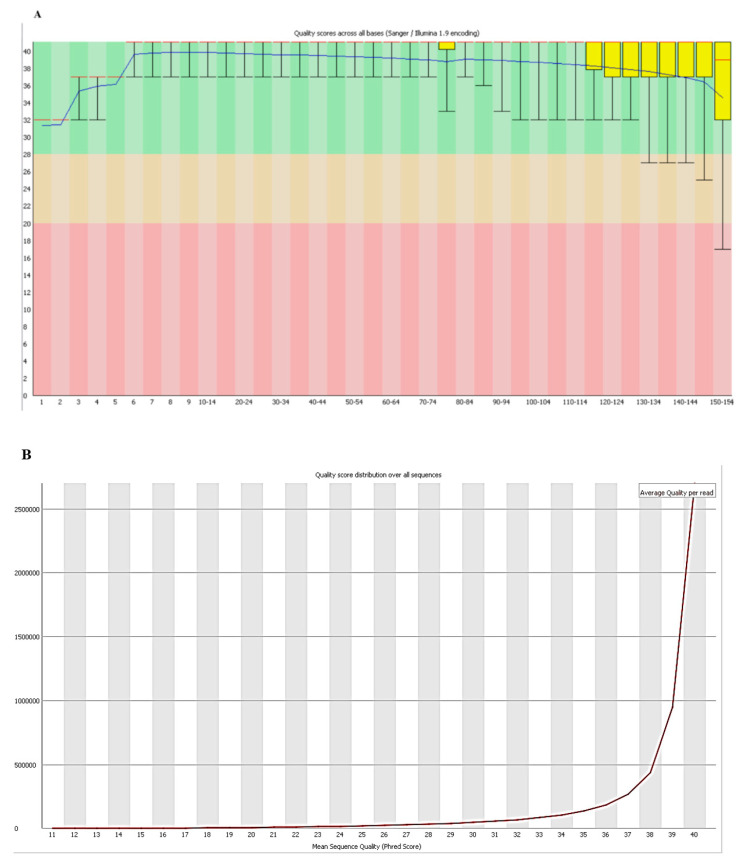
Quality assessment of reads. (**A**) Per base sequence quality. (**B**) Per base quality score. (**C**) Per sequence CG-Content. (**D**) Sequence length distribution.

**Figure 2 molecules-27-05399-f002:**
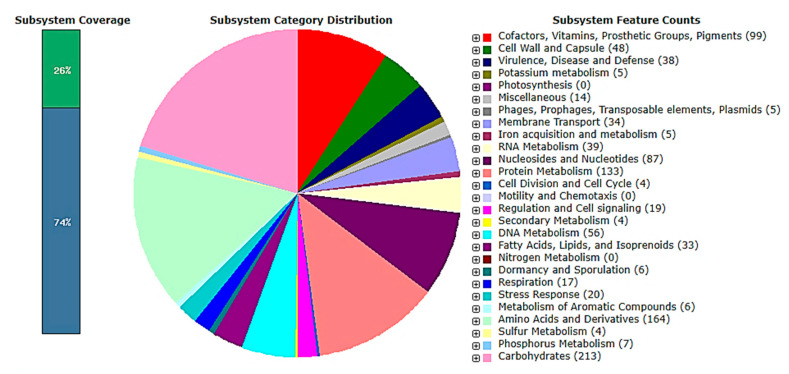
Subsystem coverage and distribution of *L. plantarum* 13-3 genome by RAST.

**Figure 3 molecules-27-05399-f003:**
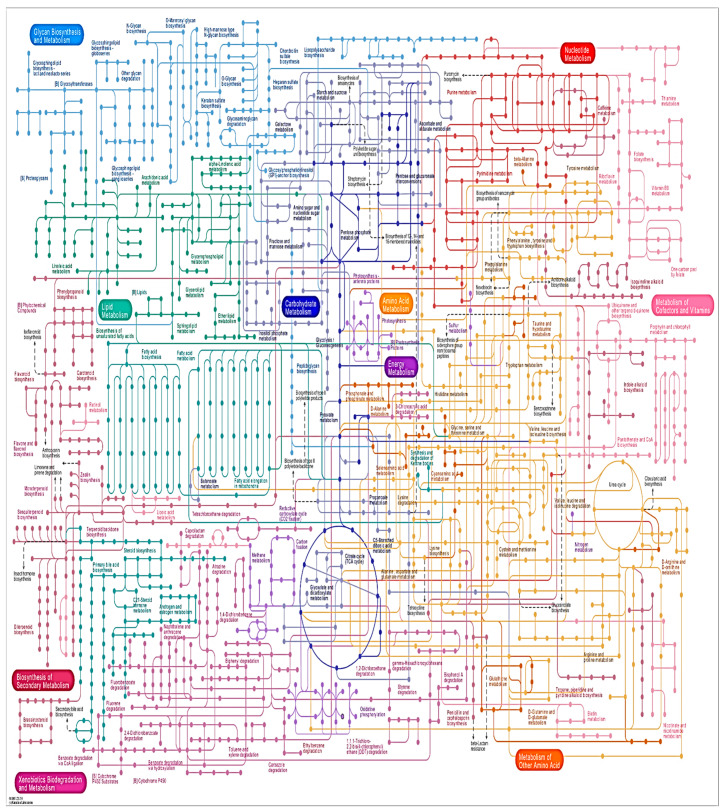
KEGG pathway of carbohydrate synthesis and secondary metabolites.

**Figure 4 molecules-27-05399-f004:**
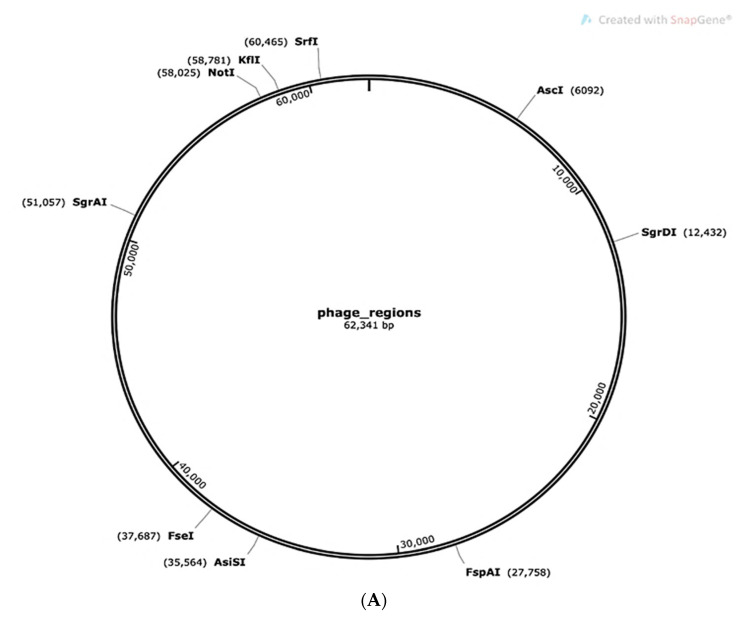
Phage site prediction by PHASTER. (**A**) Complete genome prediction of *L. plantarum* 13-3 showing the phage regions in the *lactobacillus plantarum* genome. (**B**) Expanded view of genome showing phage sites.

**Figure 5 molecules-27-05399-f005:**
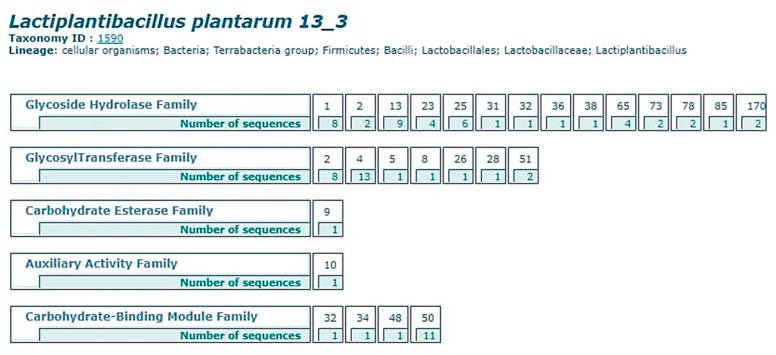
Carbohydrate-active enzymes prediction by CAZy database showing the prediction of the glycoside Hydrolase family, glycosyltransferase family, carbohydrate esterase family, auxiliary esterase family and carbohydrate binding module family.

**Figure 6 molecules-27-05399-f006:**
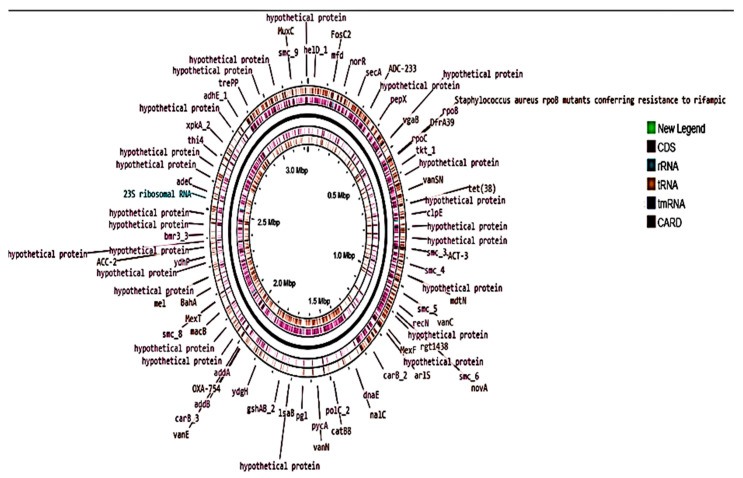
Graphical representation of functionally annotated *L. plantarum* 13-3 genome via GC-Viewer.

**Figure 7 molecules-27-05399-f007:**
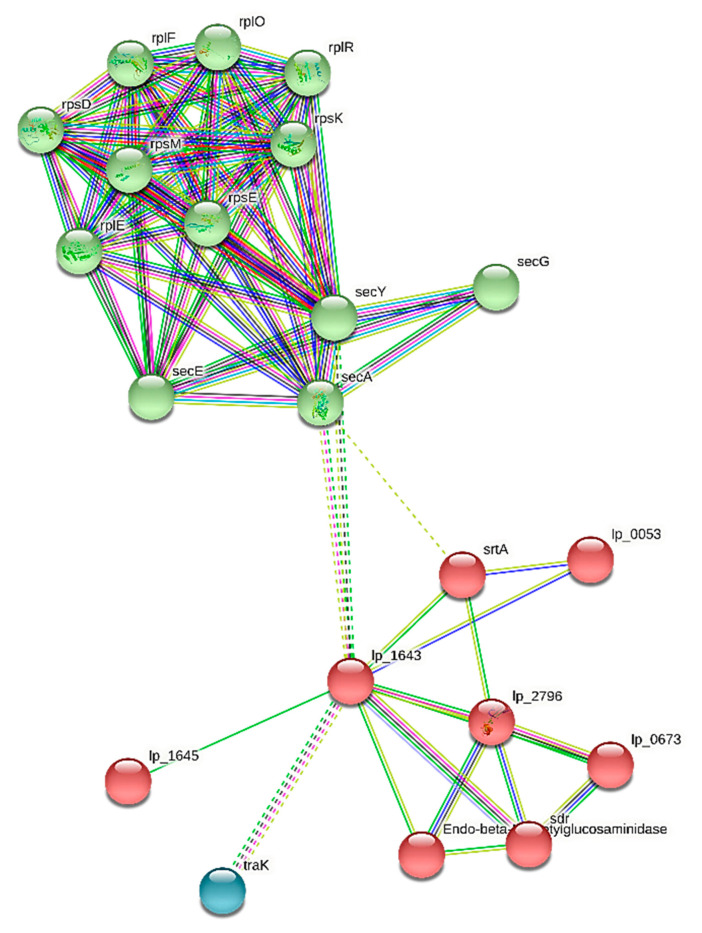
Protein–protein interaction of functionally annotated components of *L. plantarum* 13-3 genome.

**Figure 8 molecules-27-05399-f008:**
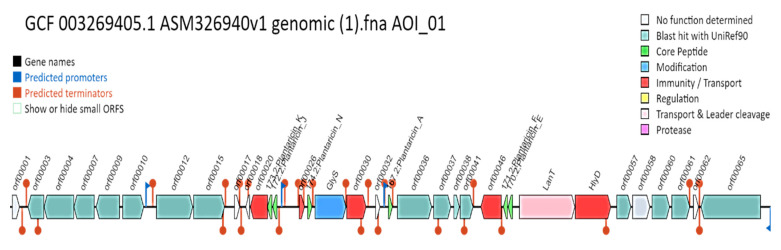
Linear graphical representation of *L. plantarum* 13-3 genome through BAGEL4 showing its non-pathogenic components, i.e., core peptides (green), immunity boosters (red), transportation components (pink), regulatory factors (yellow).

**Figure 9 molecules-27-05399-f009:**
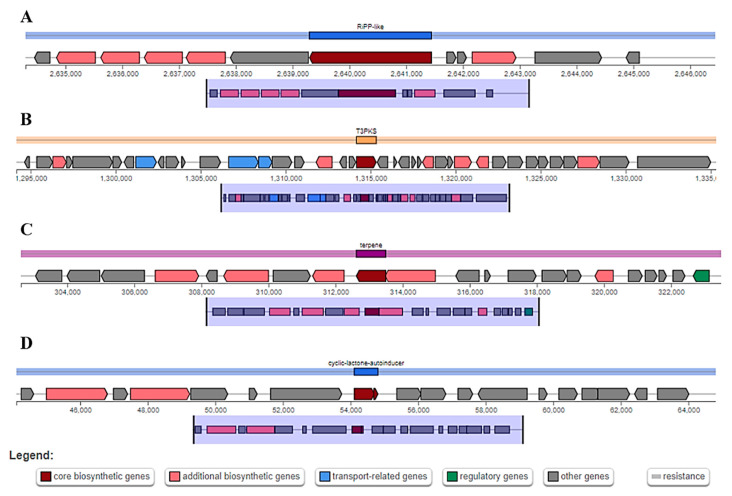
Bacteriocins and secondary metabolite-producing regions in genome of *L. plantarum* 13-3. Red (core biosynthetic genes), pink (additional biosynthetic genes), green (regulatory genes), blue (transport-related genes), grey (other genes). (**A**) cyclic lactone autoinducer. (**B**) Terpene. (**C**) T3PKS. (**D**) RiPP-like.

## Data Availability

Not applicable.

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
