# Peer review of "Functional Annotation of Lactiplantibacillus plantarum 13-3 as a Potential Starter Probiotic Involved in the Food Safety of Fermented Products"

_molecules, 2022, doi:10.3390/molecules27175399_

Round 1
Reviewer 1 Report
1-Please avoid cross reference. Insert directly the main related recent reference in any statement instead of using a second one. In the first references of the text I felt cross reference.
The main part of human diet for centuries is fermented food [1]. (1. Chilton SN, Burton JP, Reid G. Inclusion of fermented foods in food guides around the world. Nutrients, 2015; 7:390-404) In addition to those fermented foods derived from meat, milk, and plant foods have a longer shelf life as compared to fresh raw materials [2] (2. Ross RP, Morgan S, Hill C. Preservation and fermentation: Past, present and future. Int J Food Microbiol 2002; 79:3-16) and it is because of their higher water content and nutritional value. Plants and animals derived fermented foods plays a key role in the world food industry including Asia as well as western countries and the Americas (western hemisphere) [3] (3. Soemarie YB, Milanda T, Barliana MI. Fermented foods as probiotics: A review. J Adv Pharm Technol Res 2021; 12:335-9). These fermented foods contain nutrients with great potential in maintaining health and preventing diseases but also undergo changes in taste, texture, decreased toxicity, and cooking time [4-5]. (4. Kabak B, Dobson AD. An introduction to the traditional fermented foods and beverages of Turkey. Crit Rev Food Sci Nutr, 2011; 51:248-60. & 5. RolleR, Satin M. Basic requirements for the transfer of fermentation technologies to developing countries. Int J Food Microbiol, 2002; 75:181-7) Initially, lactic acid bacteria (LAB) were isolated from fermented foods, and they are the most suitable candidate for enhancing fermentation in 60 terms of product safety and to have the capability to spontaneously adapt well during the 61 fermentation process [3] (3. Soemarie YB, Milanda T, Barliana MI. Fermented foods as probiotics: A review. J Adv Pharm Technol Res 2021; 12:335-9).
2- Please insert the references in the exact place after or even within a statement, instead of leaving 7 articles in the end of paragraph:
LAB is known to inhibit diverse environments such as carbohydrate rich food plants, mucosa and intestinal environments of human and animals [6-12].
3-Recent studies suggest some metabolites of probiotics e.g. histamine and D-lactate may negatively impact host health. Some probiotic products have been suspected of inducing D-lactic-acidosis; an illness associated with neurocognitive symptoms such as ataxia
What is your idea about this problem?
4-Following sentences need reference.
The human stomach related framework contains roughly 400 different bacterial species and its overflow contrasts between people (REFERENCE). Among them few probiotic Lactobacillus species in particular, Lactobacillus acidophilus, Lactobacillus pentosus, Lactobacillus brevis, Lactobacillus lactis, Lactobacillus amylovorus, Lactobacillus casei, Lactobacillus bulgaricus, Lactobacillus fermentum, Lactiplantibacillus plantarum and Lacto- bacillus rhamnosus explicitly produce extracellular proteins, exopolysaccharides, bacteriocins and lipoteichoic acids which impact the wellbeing and physiology of the host by communicating with the epithelial cells and improve the host resistant framework [6-8]. (PLEASE DISTRIBUTE REFERENCES IN TEXT).
5-Check abbreviation rules and use them whenever is necessary. In some cases repeated words in close sentences and even 1 sentence suffer reader.
Among them few probiotic Lactobacillus species in particular, Lactobacillus acidophilus, Lactobacillus pentosus, Lactobacillus brevis, Lactobacillus lactis, Lactobacillus amylovorus, Lactobacillus casei, Lactobacillus bulgaricus, Lactobacillus fermentum, Lactiplantibacillus plantarum and Lacto- bacillus rhamnosus.
Could change to
Among them few probiotic Lactobacillus species in particular, Lactobacillus acidophilus (L.), L. pentosus, L. brevis, L. lactis, L. amylovorus, L. casei, L. bulgaricus, L. fermentum, L. plantarum and L. rhamnosus
6-Deletet some old fashion descriptions from the text which has changed introduction to a learning text instead of professional research note. All readers know something about probiotics and there is no need to add a long paragraph (from Line 76 to 90) to their properties.
7-Use new names of Species based on Berg's Classification
Lacticaseibacillus casei
Lacticaseibacillus rhamnosus
Lactiplantibacillus plantarum
Ligilactobacillus salivarius
Limosilactobacillus fermentum
Limosilactobacillus reuteri
8- Describe gap of research and point to the unsolved problems of the previous research. In other words, what is scientific contribution of this research?
9- Reduce similarity base on attached ithenticate analysis.
10- The resolution of Figures are very low, they all need to be improved to at least 300 dpi.
11-Replace following reference with a new one:
D.M. Lilly and R.H. Stillwell. Probiotics: Growth promoting factors produced by microorganisms. Science, 1965, 147, pp. 747-431 748
12-Add some related references. I recommend pointing to effect of process and storage condition to probiotics viability. In this regard, the following references could be studied. Just for instance:
Incorporation of probiotics in several foods are reported e.g. cornflakes 3, pomegranate juice 4, Doogh 5, cheese 6, fermented drink 7, yogurt 8-12, fermented milks 13, grape drink (13+), soya milk chocolate (13++) etc. Probiotics posse many health beneficial intrinsic properties for their host. Also they can produce many useful metabolites during their growth and metabolism e.g. production of bioactive compounds 14, conjugated linolenic acid 15, propionic acid 16 etc. Recently, reduction of oxidative stress and inflammatory factors (17, 18) removal of toxins and heavy metals (19) are reported for these amazing microorganisms. To stimulate probiotic growth prebiotics are administered (20-22). Increasing knowledge on human intestinal microbiota and microbiota development enables the design of new more specific and hitherto unknown probiotics and prebiotics (23).
-The Impact of Inoculation Rate and Order on Physicochemical, Microstructural and Sensory Attributes of Probiotic Doogh, Iranian Journal of Pharmaceutical Research, 2013, 12(4): 917-924
-The Effect of Homogenization Pressure and Stages on the Amounts of Lactic and Acetic Acids of Probiotic Yoghurt, Applied Food Biotechnology, 2014, 1(2), 25-29.
-Shirin Malganji, Sara Sohrabvandi, Mahshid Jahadi, Ameneh Nematollahi, Bahareh Sarmadi, Effect of Refrigerated Storage on Sensory Properties and Viability of Probiotic in Grape Drink, APPLIED FOOD BIOTECHNOLOGY, 2016, 3 (1): 59-62
-Effects of Probiotic Cells on the Mechanical and Antibacterial Properties of Sodium-Caseinate Films
-Carlos Gómez Gallego, Seppo Salminen, Novel Probiotics and Prebiotics: How Can They Help in Human Gut Microbiota Dysbiosis? APPLIED FOOD BIOTECHNOLOGY, 2016, 3 (2): 72-81

Author Response
Dear Ms. Joyce Liu,
Assistant Editor
MDPI Molecules,
Dear Madam,
Please find the attached response to reviewer comments on our research article manuscript entitled “Functional annotation of Lactiplantibacillus plantarum 13-3, as a potential starter probiotic involved in food safety of the fermented products.with manuscript id” “Molecules-1843883”
Reviewer 1
Comments and Suggestions for Authors
1-Please avoid cross reference. Insert directly the main related recent reference in any statement instead of using a second one. In the first references of the text, I felt cross reference.
The main part of human diet for centuries is fermented food [1]. (1. Chilton SN, Burton JP, Reid G. Inclusion of fermented foods in food guides around the world. Nutrients, 2015; 7:390-404) In addition to those fermented foods derived from meat, milk, and plant foods have a longer shelf life as compared to fresh raw materials [2] (2. Ross RP, Morgan S, Hill C. Preservation, and fermentation: Past, present and future. Int J Food Microbiol 2002; 79:3-16) and it is because of their higher water content and nutritional value. Plants and animals derived fermented foods plays a key role in the world food industry including Asia as well as western countries and the Americas (western hemisphere) [3] (3. Soemarie YB, Milanda T, Barliana MI. Fermented foods as probiotics: A review. J Adv Pharm Technol Res 2021; 12:335-9). These fermented foods contain nutrients with great potential in maintaining health and preventing diseases but also undergo changes in taste, texture, decreased toxicity, and cooking time [4-5]. (4. Kabak B, Dobson AD. An introduction to the traditional fermented foods and beverages of Turkey. Crit Rev Food Sci Nutr, 2011; 51:248-60. & 5. RolleR, Satin M. Basic requirements for the transfer of fermentation technologies to developing countries. Int J Food Microbiol, 2002; 75:181-7) Initially, lactic acid bacteria (LAB) were isolated from fermented foods, and they are the most suitable candidate for enhancing fermentation in 60 terms of product safety and to have the capability to spontaneously adapt well during the 61 fermentation process [3] (3. Soemarie YB, Milanda T, Barliana MI. Fermented foods as probiotics: A review. J Adv Pharm Technol Res 2021; 12:335-9).
Author Response: Thank you very much for the comment. It has been modified and corrected in the revised manuscript and highlighted in red color. Please see revised manuscript.
2- Please insert the references in the exact place after or even within a statement, instead of leaving 7 articles in the end of paragraph:
LAB is known to inhibit diverse environments such as carbohydrate rich food plants, mucosa and intestinal environments of human and animals [6-12].
Author Response: Thank you very much for the comment. It has been modified and corrected in the revised manuscript and highlighted in red color. Please see revised manuscript.
3-Recent studies suggest some metabolites of probiotics e.g. histamine and D-lactate may negatively impact host health. Some probiotic products have been suspected of inducing D-lactic-acidosis; an illness associated with neurocognitive symptoms such as ataxia
What is your idea about this problem?
Author Response: Thank you very much for the comment. Yes, some probiotics have negative impact on host health, but they are harmful as some dosage. To reduce the negative impact of probiotics we should start with a low dose of probiotics and gradually raise to the full dosage over a few weeks to minimize the possibility of negative effects. The human body will be able to adjust the probiotics timely. As it’s a probiotic bacteria derived from Tibetan kefir grains so that’s why we mentioned this sentence.
4-Following sentences need reference.
The human stomach related framework contains roughly 400 different bacterial species and its overflow contrasts between people (REFERENCE). Among them few probiotic Lactobacillus species in particular, Lactobacillus acidophilus, Lactobacillus pentosus, Lactobacillus brevis, Lactobacillus lactis, Lactobacillus amylovorus, Lactobacillus casei, Lactobacillus bulgaricus, Lactobacillus fermentum, Lactiplantibacillus plantarum and Lacto- bacillus rhamnosus explicitly produce extracellular proteins, exopolysaccharides, bacteriocins and lipoteichoic acids which impact the wellbeing and physiology of the host by communicating with the epithelial cells and improve the host resistant framework [6-8]. (PLEASE DISTRIBUTE REFERENCES IN TEXT).
Author Response: Thank you very much for the comment. Distribution of references have been revised in the revised manuscript and highlighted in red color. Please see revised manuscript.
5-Check abbreviation rules and use them whenever is necessary. In some cases repeated words in close sentences and even 1 sentence suffer reader.
Among them few probiotic Lactobacillus species in particular, Lactobacillus acidophilus, Lactobacillus pentosus, Lactobacillus brevis, Lactobacillus lactis, Lactobacillus amylovorus, Lactobacillus casei, Lactobacillus bulgaricus, Lactobacillus fermentum, Lactiplantibacillus plantarum and Lacto- bacillus rhamnosus.
Could change to
Among them few probiotic Lactobacillus species in particular, Lactobacillus acidophilus (L.), L. pentosus, L. brevis, L. lactis, L. amylovorus, L. casei, L. bulgaricus, L. fermentum, L. plantarum and L. rhamnosus
Author Response: Thank you very much for the comment. It has been modified and corrected accordingly in the revised manuscript and highlighted in red color. Please see revised manuscript.
6-Deletet some old fashion descriptions from the text which has changed introduction to a learning text instead of professional research note. All readers know something about probiotics and there is no need to add a long paragraph (from Line 76 to 90) to their properties.
Author Response: Thank you very much for the comment. The paragraph has been removed as per suggestion from the revised manuscript. Please see revised manuscript.
7-Use new names of Species based on Berg's Classification
Lacticaseibacillus casei
Lacticaseibacillus rhamnosus
Lactiplantibacillus plantarum
Ligilactobacillus salivarius
Limosilactobacillus fermentum
Limosilactobacillus reuteri
Author Response: Thank you very much for the comment. All the species names have been revised accordingly.
8- Describe gap of research and point to the unsolved problems of the previous research. In other words, what is scientific contribution of this research?
Response: Thank you very much for the comment. The current research contributes to not only the characterization of novel probiotics but also helps in determining the bacteriocins that are effective on pathogenic muti-drug resistant bacteria (mentioned in Table S6 and S10) that has a beneficial impact on the human health as compared to the previously used probiotics producing strains. Also the previously probiotics producing strains are involved in antibiotics resistant gene transfer to pathogenic bacteria, yet the use of this novel strain i.e., Lactobacillus plantrum 13-3 strain will help scientific community to overcome this gene transfer.
9- Reduce similarity base on attached ithenticate analysis.
Author Response: Thank you very much for the comment. The similarity index has been minimized.
10- The resolution of Figures are very low, they all need to be improved to at least 300 dpi.
Author Response: Thank you very much for the comment. The figures quality has been improved in the revised manuscript. Please see revised manuscript.
11-Replace following reference with a new one:
D.M. Lilly and R.H. Stillwell. Probiotics: Growth promoting factors produced by microorganisms. Science, 1965, 147, pp. 747-431 748
Author Response: Thank you very much for the comment. The reference along with its associated text have been removed as per your previous comment from the revised manuscript.
12-Add some related references. I recommend pointing to effect of process and storage condition to probiotics viability. In this regard, the following references could be studied. Just for instance:
Incorporation of probiotics in several foods are reported e.g. cornflakes 3, pomegranate juice 4, Doogh 5, cheese 6, fermented drink 7, yogurt 8-12, fermented milks 13, grape drink (13+), soya milk chocolate (13++) etc. Probiotics posse many health beneficial intrinsic properties for their host. Also they can produce many useful metabolites during their growth and metabolism e.g. production of bioactive compounds 14, conjugated linolenic acid 15, propionic acid 16 etc. Recently, reduction of oxidative stress and inflammatory factors (17, 18) removal of toxins and heavy metals (19) are reported for these amazing microorganisms. To stimulate probiotic growth prebiotics are administered (20-22). Increasing knowledge on human intestinal microbiota and microbiota development enables the design of new more specific and hitherto unknown probiotics and prebiotics (23).
-The Impact of Inoculation Rate and Order on Physicochemical, Microstructural and Sensory Attributes of Probiotic Doogh, Iranian Journal of Pharmaceutical Research, 2013, 12(4): 917-924
-The Effect of Homogenization Pressure and Stages on the Amounts of Lactic and Acetic Acids of Probiotic Yoghurt, Applied Food Biotechnology, 2014, 1(2), 25-29.
-Shirin Malganji, Sara Sohrabvandi, Mahshid Jahadi, Ameneh Nematollahi, Bahareh Sarmadi, Effect of Refrigerated Storage on Sensory Properties and Viability of Probiotic in Grape Drink, APPLIED FOOD BIOTECHNOLOGY, 2016, 3 (1): 59-62
-Effects of Probiotic Cells on the Mechanical and Antibacterial Properties of Sodium-Caseinate Films
-Carlos Gómez Gallego, Seppo Salminen, Novel Probiotics and Prebiotics: How Can They Help in Human Gut Microbiota Dysbiosis? APPLIED FOOD BIOTECHNOLOGY, 2016, 3 (2): 72-81
Author Response: Thank you very much for the detailed comment. The said references have been added along with the paragraph to the revised manuscript and highlighted in red color. Please see revised manuscript.
Regards
Prof Dr Yang Zhennai
Professor
School of Food Science & Health
Beijing Technology & Business University Beijing
Reviewer 2 Report
Tariq Aziz et al. submitted to Molecules an article focusing on L. plantarum 13-3, as a potential starter probiotic involved in the food safety of fermented products. The manuscript appears adequately structured according to the scientific method and lends itself to a fluid and interesting reading.
Dealing with health disciplines adhering to Food Safety and Public Health, I am not able to enter into the merits of the aspects concerning DNA extraction and genomic investigation, but I believe that further efforts can be made to refine the article and make it more appealing to the expert reader:
L 108: please write in full Aspergillus niger
LL 124-132: it is not correct to add these sentences in the introduction, please move them to discussions or conclusions.
Materials and Methods section: please specify the time period in which the study was conducted
The quality of figure 1 is not adequate as it is totally grainy, please provide better quality images. Likewise for figure 4.
L 329: “generally regarded as safe”, I suppose this aspect deserves attention and in-depth study, since the title specifically refers to the terminology “food safety”. Does “as safe” relate to healthy subjects or is it a microorganism with potential pathogenesis capabilities if the individual is in certain conditions? This aspect is to be explored, also taking into account specific literature data. Furthermore, I suppose it is important to consider the way of transport and storage (eg maintenance of the cold chain) that can affect live and viable microorganisms and, furthermore, have shelf-life tests been carried out?
L 357: please better detail what you mean by “food safety agent”. It refers to sector regulations (eg EC Reg. 2073/05?) or, in any case, it is necessary to explain in detail how it is possible to provide such a statement with certainty.
Discussion section: please make clear any limitations regarding this study.
Reference 47: I checked in Pubmed and the authors are “Devi SM, Halami PM”, so there is probably a mistake in “&”
The references are correctly cited, according to the Instructions for the Authors of this Journal.
Some grammatical and syntax variations are necessary, please take the opportunity to have the work at least reread by a native English speaker colleague.
Thank you for your efforts in perfecting this important article.
Author Response
Dear Ms. Joyce Liu,
Assistant Editor
MDPI Molecules,
Dear Madam,
Please find the attached response to reviewer comments on our research article manuscript entitled “Functional annotation of Lactiplantibacillus plantarum 13-3, as a potential starter probiotic involved in food safety of the fermented products. With manuscript id” “Molecules-1843883”
Reviewer 2
Comments and Suggestions for Authors
Tariq Aziz et al. submitted to Molecules an article focusing on L. plantarum 13-3, as a potential starter probiotic involved in the food safety of fermented products. The manuscript appears adequately structured according to the scientific method and lends itself to a fluid and interesting reading.
AR: Thank you very much for your comments and appreciation. Means a lot to us and its encouragement for us.
Dealing with health disciplines adhering to Food Safety and Public Health, I am not able to enter into the merits of the aspects concerning DNA extraction and genomic investigation, but I believe that further efforts can be made to refine the article and make it more appealing to the expert reader:
AR: Thank you very much for your comments and appreciation. Means a lot to us and its encouragement for us.
L 108: please write in full Aspergillus niger
AR: Thank you very much for your comment. It has been written as Aspergillus niger and highlighted in yellow color in the revised manuscript. Please see revised manuscript.
LL 124-132: it is not correct to add these sentences in the introduction, please move them to discussions or conclusions.
AR: Thank you very much for your comment. It has been moved to conclusion in the revised manuscript and highlighted in yellow color. Please see revised manuscript.
Materials and Methods section: please specify the time period in which the study was conducted.
AR: Thank you very much for your comments. These strains were isolated in 2015 and were further characterized in 2016. It was further studied in 2016-2020. Tariq Aziz (the Ist author) worked on it in several activities during his PhD studies from 2016-2020. Lactiplantibacillus plantarum 13-3 whole genome sequencing was done in 2019 and some articles on this strain was published by Tariq Aziz in 2019-2020.
The quality of figure 1 is not adequate as it is totally grainy, please provide better quality images. Likewise for figure 4.
AR: Thank you very much for the comment. Figure quality improved.
L 329: “generally regarded as safe”, I suppose this aspect deserves attention and in-depth study, since the title specifically refers to the terminology “food safety”. Does “as safe” relate to healthy subjects or is it a microorganism with potential pathogenesis capabilities if the individual is in certain conditions? This aspect is to be explored, also taking into account specific literature data. Furthermore, I suppose it is important to consider the way of transport and storage (eg maintenance of the cold chain) that can affect live and viable microorganisms and, furthermore, have shelf-life tests been carried out?.
AR: Thank you very much for your comment. This strain was deeply studied along with some other strains in terms of food safety, GRAS, its storage and transportation and how much important is this strain in dairy industry. A research article was published by the following authors Yang Ya-wei ,Zhao Ai-mei,Wang Ji,Yang Zhen-nai. Separation and screening for exopolysaccharide-producing Lactic acid bacteria from milk tofu and their potential probiotic properties research. China Dairy Research Journal, 2015, Vol 43, Issue 12. www.chinadairy.net. Furthermore, this strain has the capability to transform linoleic acid into various fatty acid metabolites. An article entitled “CONVERSION OF LINOLEIC ACID TO DIFFERENT FATTY ACID METABOLITES BY Lactobacillus Plantarum 13-3 AND IN SILICO CHARACTERIZATION OF THE PROMINENT REACTIONS” was published by Tariq Aziz et al in 2020, in Journal of the Chilean Chemical Society , J. Chil. Chem. Soc., 65, N°3 (2020).
L 357: please better detail what you mean by “food safety agent”. It refers to sector regulations (eg EC Reg. 2073/05?) or, in any case, it is necessary to explain in detail how it is possible to provide such a statement with certainty.
AR: Thank you very much for the comment. Added detail in lines (343-345).
Discussion section: please make clear any limitations regarding this study.
AR: Thank you very much for the comment. The main limitation of this study is that the computationally determined bacteriocins and probiotics of molecularly characterized strain Lactobacillus plantrum 13-3 are not analyzed in wet lab.
Reference 47: I checked in Pubmed and the authors are “Devi SM, Halami PM”, so there is probably a mistake in “&”
AR: Thank you very much for the correction. Yeah, the correction reference is Devi SM & Halami PM. Genetic Variation of pln Loci Among Probiotic Lactobacillus plantarum Group Strains with Antioxidant and Cholesterol-Lowering Ability. Probiotics Antimicrob Proteins. 2019, 11(1):11-22. It has been corrected in the revised manuscript.
The references are correctly cited, according to the Instructions for the Authors of this Journal.
AR: Thank you very much for your comments and appreciation. Means a lot to us and its encouragement for us.
Some grammatical and syntax variations are necessary, please take the opportunity to have the work at least reread by a native English speaker colleague.
AR: Thank you very much for your comments. English language editing has be done through a native speaker in the revised manuscript.
Thank you for your efforts in perfecting this important article.
AR: Thank you very much for your comments and appreciation. Means a lot to us and its encouragement for us.
Regards
Prof Dr Yang Zhennai
Professor
School of Food Science & Health
Beijing Technology & Business University Beijing
Round 2
Reviewer 1 Report
The manuscript has been improved significantly. But the figures are not still with high resolution.
Best
Author Response
Dear Ms. Joyce Liu,
Assistant Editor
MDPI Molecules,
Dear Madam,
Please find the attached response to reviewer comments on our research article manuscript entitled “Functional annotation of Lactiplantibacillus plantarum 13-3, as a potential starter probiotic involved in food safety of the fermented products. With manuscript id” “Molecules-1843883”
Reviewer 1 (Round 2)
Comments and Suggestions for Authors
The manuscript has been improved significantly. But the figures are not still with high resolution.
Best
AR: Thank you very much for your comments. The figures quality has been improved in the revised manuscript (round 2). Please see the revised manuscript.
Regards
Prof Dr Yang Zhennai
Professor
School of Food Science & Health
Beijing Technology & Business University Beijing
Reviewer 2 Report
The answers provided are partially exhaustive, having been indicated only as a response to the referee and not expressly included in the manuscript.
L 126: “as previously described by […]”: it seems that something is missing.
I believe it is right to include in the manuscript the time period in which the study has been conducted, as it was clarified in the response to the referee. Likewise regarding the study limits, which must be indicated in the article using the terminology "limits".
Author Response
Dear Ms. Joyce Liu,
Assistant Editor
MDPI Molecules,
Dear Madam,
Please find the attached response to reviewer comments on our research article manuscript entitled “Functional annotation of Lactiplantibacillus plantarum 13-3, as a potential starter probiotic involved in food safety of the fermented products. With manuscript id” “Molecules-1843883”
Reviewer 2 (Round 2)
Comments and Suggestions for Authors
The answers provided are partially exhaustive, having been indicated only as a response to the referee and not expressly included in the manuscript.
L 126: “as previously described by […]”: it seems that something is missing.
AR: Thank you very much for your comments. It has been modified and correct as as previously described by (Aziz, T., et al., 2022; 2021;2020a;2020b & 2020c; Jian et al;2017 & 2020; Zhang et al., 2022 & 2020; Yunyun et al 2018; Wang et al., 2015) [32-41]. It has been highlighted in yellow color in the revised manuscript. Please see revised manuscript.
I believe it is right to include in the manuscript the time period in which the study has been conducted, as it was clarified in the response to the referee. Likewise regarding the study limits, which must be indicated in the article using the terminology "limits".
AR: Thank you very much for your comments. As responded to your previous comments we mentioned that these strains were isolated in 2015 and were further characterized in 2016. It was further studied in 2016-2020. Tariq Aziz (the Ist author) worked on it in several activities during his PhD studies from 2016-2020. Lactiplantibacillus plantarum 13-3 whole genome sequencing was done in 2019 and some articles on this strain was published by Tariq Aziz in 2019-2020. Now the year has been mentioned as “The strain Lactiplantibacillus plantarum 13-3 (L. plantarum 13-3) was previously isolated from Tibetan Kefir grains in 2015. Further periods are provided in lines (143, 146, 150, 152, 154-155, 157, 158, 161, 164, 165, 167, 169, 171, 173, 176, 180, and 183).
Regards
Prof Dr Yang Zhennai
Professor
School of Food Science & Health
Beijing Technology & Business University Beijing